# Build your own cell: Diffusion Models for Multichannel 3D Microscopy Image Generation

## Abstract

Three-dimensional (3D) cellular morphology is a critical indicator of cellular function, disease states, and drug responses. However, capturing and interpreting the complex relationships between cell shape, treatment conditions, and their biological implications remains a challenge. To address this, we present "Build Your Own Cell" (BYOC), a multichannel 3D generative framework that combines vector quantisation and diffusion models to synthesise biologically realistic 3D cell structures. BYOC captures intricate morphological changes induced by different drug treatments, enabling high-throughput in silico simulations and screening of cell shapes in response to varied conditions. This novel framework represents a significant step towards accelerating pre-clinical drug development by synthesising high-resolution, biologically realistic 3D cells, potentially reducing reliance on labour-intensive experimental studies. By ensuring phenotypic consistency between cell and nucleus volumes through joint modelling, BYOC provides high-fidelity reconstructions that could facilitate downstream analyses, including drug efficacy evaluation and mechanistic studies. Our project repository is at https://anonymous.4open.science/r/ICLR_BYOC/README.md.

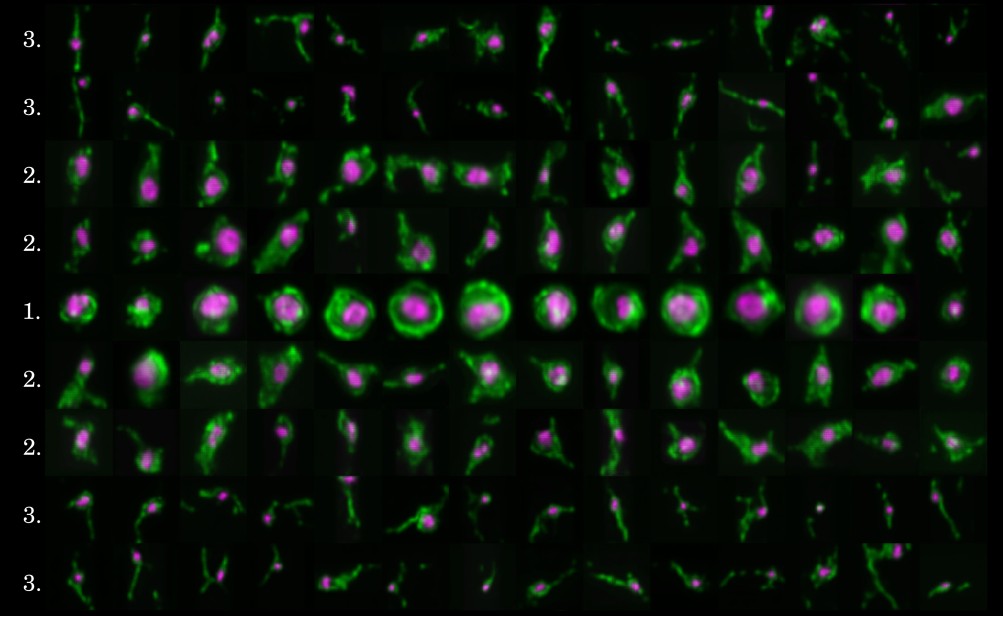

Figure 1: BYOC-generated 3D cellular structures, showcasing a continuous transformation of cell and nucleus morphologies under distinct drug treatments; (1) Nocodazole, (2) Binimetinib and (3) Blebbistatin. This visual highlights the adaptability of our generative model in capturing biologically relevant morphological diversity.

## 1 INTRODUCTION

Generative models have made remarkable strides in achieving realistic synthetic outputs (Rombach et al., 2022; Ramesh et al., 2022; 2021), but are still far from presenting a convincing understanding of complex biological structures. As these methods expand into safety-critical domains, such as drug discovery and clinical decision-making, the demand for generated samples that are not only realistic but also phenotypically accurate has grown. Medical imaging, in particular, requires 3D volumetric synthesis that can faithfully capture intricate dimension- and channel-specific features essential for precise analysis. A lack of inter- and intra-channel consistency in synthetic biological structures can lead to erroneous conclusions, affecting both diagnostic accuracy and treatment evaluation. Current high-resolution 3D synthesis methods are often not tailored to model the associations and relationships between channels that together define a biological structure. In contrast, machines that *do* attend to each channel in a unified framework hold the potential to synthesise structures that more closely resemble natural biological forms. In this paper, we address these challenges by introducing a novel framework for multichannel 3D volumetric generation, focusing on the simultaneous synthesis of cell and nucleus structures in response to different drug treatments.

Within the broader family of generative models, diffusion models have emerged as compelling tools for image synthesis. These models can operate in an unconditional framework, where realistic outputs are synthesised from random Gaussian noise (Ho et al., 2020), or in a conditional framework, where specific tasks guide the generative process. Notable examples of the latter include text-to-image generation (Zhang et al., 2023; Ramesh et al., 2022; Saharia et al., 2022), image-to-video translation (Ni et al., 2023), and 2D-to-3D reconstruction (Shi et al., 2024; Poole et al., 2022). Other works have explored multimodal diffusion modelling (Ruan et al., 2023) for multi-modality generation. Despite these advancements, the synthesis of full 3D volumetric images remains relatively underexplored, particularly when compared to the progress made in point cloud (zeng et al., 2022) and mesh-based (Liu et al., 2023) 3D representations. Addressing the literature gap, the works of Khader et al. (2023), Tudosiu et al. (2024), and Sun et al. (2022) investigated the generation of high-resolution 3D volumetric images, with a focus on MRI and CT scans. These works exemplify the potential of generative models in medical imaging, highlighting their capability to synthesise detailed, high-fidelity 3D structures. These contributions pave the way for broader applications of medically-derived volumetric generation but fall short when considering medical samples that are comprised of multiple channels.

A significant area where multichannel volumetric generation holds promise is in the synthesis of cellular structures. Cellular morphology encodes biological information such as cellular function and state (Bakal et al., 2007; Lomakin et al., 2020), providing insights into processes such as disease progression and drug response. Traditionally, studying these phenomena has relied heavily on labour-intensive lab experimentation and 3D imaging, limiting the scale and efficiency of such investigations. The ability to generate biologically realistic 3D cellular structures represents a foundational first step toward virtual screening pipelines, enabling high-throughput analysis of drug-induced morphological changes. By replicating the intricate features of cellular architectures, this approach facilitates new opportunities for understanding the effects of therapeutic interventions efficiently and at scale.

However, achieving realistic synthesis of multichannel 3D cellular volumes introduces unique technical challenges. Fluorescence microscopy datasets often comprise multiple channels (Chandrasekaran et al., 2023; Chen et al., 2023), each encoding distinct but related biological features, comprising multiple organelles and cellular compartments. In our work, we specifically focus on the synthesis of 3D cellular structures comprising two key channels: the cell and nucleus, which play central roles in encoding cellular state and function. The relationship between the cell and nucleus is biologically intertwined, necessitating a generative framework that captures both inter-channel dependencies and intra-channel consistency. In fluorescence microscopy, additional challenges arise from high-resolution single-cell data being both high-dimensional and inconsistent in size, referring to the varying dimensions of the images themselves. These variations stem from biological heterogeneity, making accurate synthesis particularly demanding. To address these challenges, **our key contributions are as follows:**

1. We propose the first 3D fluorescence cell generative model, introducing a library of codebooks designed to independently process each biological channel (cell and nucleus) while

simultaneously learning the intricate interdependencies between them, ensuring biologically accurate synthesis.

2. We adopt multimodal diffusion modelling to synthesise cell and nucleus volumes in parallel, preserving structural consistency and spatial relationships across channels.

## 2 RELATED WORK

**3D Synthesis.** Generative models, unlike discriminative frameworks that prioritise predictive accuracy and can overlook task-irrelevant details, model the underlying distribution explicitly to produce realistic and convincing outputs. As an example, a discriminative framework may be trained to classify drug-treated cells, but they may ignore subtle morphological phenotypes that do not influence classification accuracy. Generative models, in contrast, must capture the finer details that align with the complete input distribution to synthesise biological realism. This makes 3D generative tasks particularly challenging, as the synthesis must capture fine-grained details across all spatial dimensions and channels. Notably, the advent of 3D reconstruction has been popularised with methods that alter the representation of inputs to facilitate more tractable generative pipelines. Early approaches primarily focused on point clouds (zeng et al., 2022; Charles et al., 2017; Lassner & Zollhofer, 2021), voxel grids (Ren et al., 2024; Schwarz et al., 2022; Nguyen-Phuoc et al., 2019), neural fields (Xie et al., 2022), and mesh-based representations (Liu et al., 2023; Gao et al., 2022). Each of these methods offers unique benefits in terms of processing 3D data representations, laying the groundwork for more efficient 3D generation pipelines.

To facilitate the processing of these diverse data representations, techniques such as Denoising Diffusion Probabilistic Models (DDPMs) (Ho et al., 2020), Variational Autoencoders (VAEs) (Kingma & Welling, 2022), autoregressive models (van den Oord et al., 2019), and Generative Adversarial Networks (GANs) (Goodfellow et al., 2014; Wu et al., 2016) have emerged as key players in generative modelling. Among these, DDPMs have demonstrated particularly promising results for 3D generation. Unlike GANs, which often struggle with generating coherent latent representations, DDPMs are able to synthesise detailed 3D volumes from latent inputs with greater accuracy. Additionally, they produce higher-quality outputs compared to VAEs, which are often limited by blurry reconstructions (Anciukevičius et al., 2024).

**Diffusion Models for High-dimensional data.** Generative models often carry the drawback of computational inefficiency, especially when encountered with high-dimensional data. A step toward universality and controllability in generative frameworks involves enabling architectures to better process and represent such complex data, ultimately enabling more efficient and flexible generation of complex structures. Mitigating this drawback, recent literature has demonstrated the effectiveness of downsampling high-dimensional continuous voxel representations into vector quantised latent spaces (Esser et al., 2021). These quantised representations often facilitate GAN- and VAE-based architectures, enabling high-quality synthesis, particularly in medical imaging domains (Khader et al., 2023; Tudosiu et al., 2024; Sun et al., 2022). Latent compression helps overcome the computational challenges of high-dimensional datasets while preserving key features for realistic 3D generation. However, methods that focus on storing latent representations across channels remain largely underexplored.

**Multimodal synthesis.** Building on the success of single-modal generative models, multimodal generative modelling extends these capabilities by leveraging a "joint representation" across multiple data sources, commonly referred to as a "general-purpose prior." This joint representation allows for richer and more cohesive generation across various domains, where the goal is to ensure that the underlying characteristics of each modality are maintained while capturing the relationships between them. A strong multimodal representation can be decoded into multiple perturbations while retaining the integrity of the original multimodal inputs. Representation learning has been employed to achieve this objective, with techniques like VAEs being used to enforce consistency across modalities (Bengio et al., 2013). A notable example is *MM-Diffusion* (Ruan et al., 2023), which introduces a unified framework for joint high-fidelity audio-video synthesis. Multimodal generative approaches (Lee et al., 2018; Zhu et al., 2017) have shown significant promise and are certainly not limited to frameworks with distinct modalities. In fact, multiple colour channels within the same input can be treated as distinct modalities, extending this framework to use cases like biological imaging, where each channel captures related but distinct information.

## 3 METHODOLOGY

In this work, we aim to address the challenges of generating high-resolution, multichannel 3D cellular structures by building upon a hybrid generative framework (Khader et al., 2023) that leverages vector quantisation and denoising diffusion models. Our approach builds on the ability of autoencoders to efficiently represent complex data in a latent space and extends this representation using diffusion processes for realistic synthesis of multichannel volumetric data. By capturing the nuances of both global and local structures, particularly across multiple biological channels (cell and nucleus), we ensure that the synthesised volumes maintain high fidelity and consistency between the channels.

The core contribution of our framework consists of two components: (1) a library of vector quantised codebooks to learn distinct representations for each channel, and (2) a multichannel diffusion-based model for refining these representations. Section 3.1 introduces the construction of the library of codebooks using a VQGAN-based architecture (Esser et al., 2021), while Section 3.2 describes the incorporation of multichannel denoising diffusion models to ensure realistic and coherent generation of 3D volumes. By combining these two methodologies, we offer a solution that generates detailed 3D volumes from multichannel data, overcoming the limitations of previous single-channel generative approaches.

### 3.1 A LIBRARY OF CODEBOOKS

Initially dubbed as *taming transformers* for high-resolution image synthesis (Esser et al., 2021), the departure from individual pixel representation was proposed through a vector quantisation step. More specifically, the authors introduce a discrete *codebook* of learned representations, such that any input can be represented as a spatial collection of a subset of these codebook entries. Extending this concept to multichannel volumetric data—comprising both cell and nucleus channels—we define a *library of codebooks* as a collection of independently learned representations that encode the distinct features of each channel.

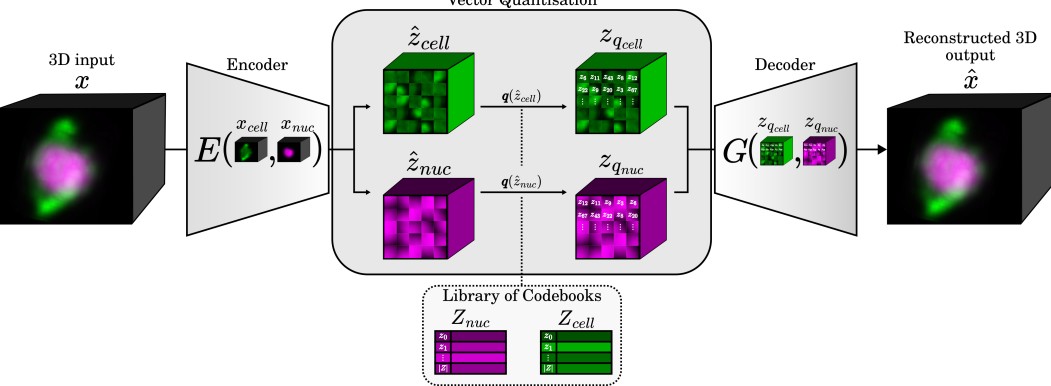

Figure 2: Depiction of the adapted multichannel VQGAN, comprising an encoding and decoding step. The library of codebooks encodes distinct spatial features for both the cell and nucleus components, which are then decoded to generate the final output.

Consider a 3D multichannel volume, $x \in \mathbb{R}^{C \times H \times W \times D}$, where $C$ represents the number of channels, $H$ the height, $W$ the width, and $D$ the depth of the volume. The volume can be decomposed into two distinct components: the cell channel, denoted as $x_{\text{cell}} \in \mathbb{R}^{1 \times H \times W \times D}$, and the nucleus channel, denoted as $x_{\text{nuc}} \in \mathbb{R}^{1 \times H \times W \times D}$. The encoded representations,

$$\hat{z}_{cell} = E(x_{nuc}) \in \mathbb{R}^{1 \times h \times w \times n_z} \text{ and } \hat{z}_{nuc} = E(x_{nuc}) \in \mathbb{R}^{1 \times h \times w \times n_v}, \tag{1}$$

where $E$ denotes the encoder and $n_z$ and $n_v$ represent the dimensionality of latent feature maps, are leveraged for representation learning (Van Den Oord et al., 2017). This process maps the inputs into separate spatial sets of codebook entries—known as quantised representations—denoted as $z_{q_{cell}}$ and $z_{q_{nuc}}$, which correspond to the downsampled spatial representations of the input volumes ($h <$

$H, w < W$, and $n_z, n_v < D$). The learned library of discrete codebooks enables the formulation of the corresponding quantised representations, where each codebook is formally defined as:

$$Z_{cell} = \{z_k\}_{k=1}^{K} \in \mathbb{R}^{1 \times n_z} \text{ and } Z_{nuc} = \{z_p\}_{p=1}^{P} \in \mathbb{R}^{1 \times n_v}, \tag{2}$$

where $K$ and $P$ denote the number of codebook entries. More precisely, obtaining the quantised representations, leveraging the learned library of discrete codebooks, is enabled through a quantisation step, denoted by $\mathbf{q}$. This operates on $\hat{z}_{ij} \in \mathbb{R}^{n_z}$ and $\hat{z}_{mn} \in \mathbb{R}^{n_v}$, and is defined as follows:

$$z_{q_{cell}} = \mathbf{q}(\hat{z}_{cell}) = \big( \arg \min_{z_k \in Z_{cell}} ||\hat{z}_{ij} - z_k|| \big), \text{and} \tag{3}$$

$$z_{q_{nuc}} = \mathbf{q}(\hat{z}_{nuc}) = \big( \arg \min_{z_p \in Z_{nuc}} ||\hat{z}_{mn} - z_p|| \big). \tag{4}$$

Equations 3 and 4, highlighting the vector quantisation, can be understood as a process whereby each vector in the unquantised representations, $z_{q_{cell}} \in \mathbb{R}^{1 \times h \times w \times n_z}$ and $z_{q_{nuc}} \in \mathbb{R}^{1 \times h \times w \times n_v}$, are replaced with the closest vector in their corresponding learned codebooks, $Z_{cell}$ and $Z_{nuc}$. After this quantisation step, the decoder uses these quantised representations to generate the final output. Formally, the generative output, $\hat{x}$, is defined as follows, where $G$ denotes the decoder:

$$\hat{x} = G(z_{q_{cell}}, z_{q_{nuc}}) = G(\mathbf{q}(E(x_{cell})), \mathbf{q}(E(x_{nuc}))). \tag{5}$$

After obtaining the generative output, $\hat{x}$, the quality and accuracy of this synthesis are guided by a set of optimisation objectives. Specifically, the learning objective of the VQGAN, as an adapted multichannel formulation (Esser et al., 2021), combines minimising a reconstruction loss, commitment loss, and discriminator loss:

$$L_{rec} = 1/2[||x_{cell} - \hat{x}_{cell}||^2 + ||x_{nuc} - \hat{x}_{nuc}||^2], \tag{6}$$

$$L_{comm} = 1/2[||sg[z_{q_{cell}}] - E(x_{cell})||_2^2 + ||sg[z_{q_{nuc}}] - E(x_{nuc})||_2^2], \tag{7}$$

$$L_{disc} = 1/2[\mathbb{E}_x(ReLU(1 - D(x)) + \mathbb{E}_{\hat{x}}(ReLU(1 - D(\hat{x}))], \tag{8}$$

where $sg$ is the stop gradient operation, and $D(x)$, $D(\hat{x})$ denote the discriminator outputs for the real and generated samples, respectively. In our adaptation, the combined channel-specific reconstruction, commitment, and discriminator losses enable the VQGAN to compress semantically rich latent representations from both cell and nucleus channels. The reconstruction loss ensures accuracy, the commitment loss maintains consistency with the quantised codebooks, and the discriminator loss promotes realism in the generated outputs. Employing a quantisation step that leverages a library of independently learned codebooks facilitates a robust framework for learning multichannel 3D representations.

## 3.2 COMPOSITION OF MULTICHANNEL VOLUMES WITH DENOISING DIFFUSION

Building on the latent representations established in Section 3.1, the next stage of our approach leverages multimodal DDPMs to generate realistic multichannel volumes. This process refines the unquantised representations of the cell and nucleus components, ensuring that both spatial and interchannel dependencies are preserved throughout the synthesis.

**Diffusion preliminaries:** DDPMs (Ho et al., 2020; Song et al., 2022) comprise a noising and de-noising iterative process. In the forward process, noise is gradually added over $T$ timesteps, transforming the input sample, $x_0$, into a latent representation that follows a unit variance normal distribution. The noisy sample, $x_t$, at each timestep, $t$, is generated according to:

$$x_t = \sqrt{\bar{\alpha}_t}x_0 + \sqrt{1 - \alpha_t}\mathbf{z}, \mathbf{z} \sim \mathcal{N}(0, \mathbf{I}), \tag{9}$$

where $\alpha_t = 1 - \beta_t$, $\bar{\alpha}_t = \Pi_{s=1}^{t}\alpha_s$, and $\beta_t$ is a predefined variance schedule. The noise schedule $\beta_t$ typically increases over time, following a cosine schedule, as proposed by (Nichol & Dhariwal, 2021). Thus, the non-parametric forward diffusion Markovian process is defined as:

$$q(x_t|x_{t-1}) = \mathcal{N}(x_t; \sqrt{1 - \beta}x_{t-1}, \beta_t\mathbf{I}). \tag{10}$$

In the reverse process, the model progressively learns to denoise the latent representation to reconstruct the original input. The iterative noise reduction process towards the original input, $x_0$, can be thought of as training a model $\theta$ to approximate "the reverse of the forward process." Specifically, the model learns $p_\theta(x_{t-1}|x_t)$ which approximates $q(x_{t-1}|x_t, x_0)$ for all timesteps $t$ and states

$x_t$. Implicitly, this approximation paramaterises Gaussian transitions, and therefore allows for the simplified formulation of the reverse process:

$$p_\theta(x_{t-1}|x_t) = \mathcal{N}(x_{t-1}; \mu_\theta(x_t, t), \sigma_\theta^2(x_t, t)), \tag{11}$$

where $\mu_\theta$ and $\sigma_\theta^2$ denote the mean and variance predicted by $\theta$.

**Multichannel Diffusion models:** With the forward and reverse diffusion processes defined in the preliminaries, we extend this framework into a multichannel perspective (Ruan et al., 2023). In the context of our BYOC pipeline, the simultaneous recovery of both cell and nucleus channels is an application of diffusion modelling in the latent space, where the high dimensionality of each channel of the data necessitates operating on compressed representations (Rombach et al., 2022).

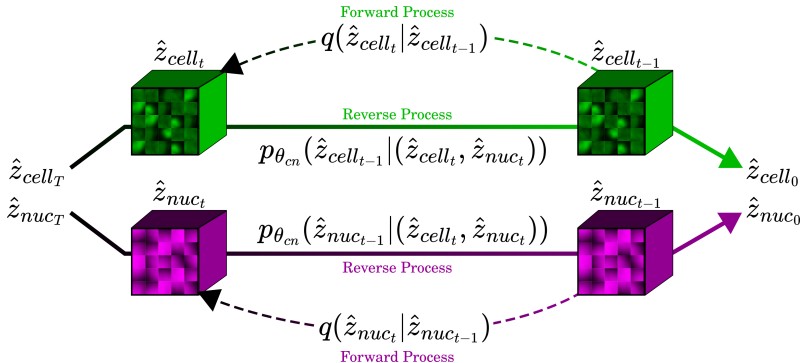

Figure 3: Illustration of the 3D diffusion process applied to channel-specific unquantised latent representations for both cell and nucleus. In the forward process, noise is added independently to each latent representation across multiple timesteps. During the reverse process, the denoising of the cell and nucleus latents occurs within a unified framework, where both channels are co-dependent, ensuring that information from one channel influences the reconstruction of the other.

Specifically, the target reconstruction is performed directly on the unquantised latent representations, $\hat{z}_{cell}$ and $\hat{z}_{nuc}$ within a unified diffusion process. Considering the unquantised cell channel latent representation, the forward process can be redefined as:

$$q(\hat{z}_{cell_t}|\hat{z}_{cell_{t-1}}) = \mathcal{N}(\hat{z}_{cell_t}; \sqrt{1-\beta_t}\hat{z}_{cell_{t-1}}, \beta_t\mathbf{I}), \tag{12}$$

where $t$ represents the diffusion timestep, ranging from $0$ to $T$. The unquantised nucleus latent representation follows an identical formulation to Equation 12, and both channels are perturbed using the same noise scheduler, $\beta$. Analogous to the implementation of *MM-Diffusion* (Ruan et al., 2023), we enforce a unified approach that approximates a multichannel, joint reverse process. The joint reverse process can be represented as a unified model enforced on the unquantised latents, $p_{\theta\hat{z}_{q_{cell}}\hat{z}_{q_{nuc}}}$, but for notational simplicity we will refer to to this reverse process as $p_{\theta_{cn}}$. Therefore, considering the unquantised cell channel latent representation, the reverse process is formulated:

$$p_{\theta_{cn}}(\hat{z}_{cell_{t-1}}|(\hat{z}_{cell_t}, \hat{z}_{nuc_t})) = \mathcal{N}(\hat{z}_{cell_{t-1}}; \mu_{\theta_{cn}}(\hat{z}_{cell_t}, \hat{z}_{nuc_t}, t)). \tag{13}$$

This suggests that, instead of independently modelling each unquantised cell and nucleus latent, the generation of the denoised channel-specific sample at timestep $t-1$ is dependent on both $z_{cell_t}$ and $z_{nuc_t}$.

**DCUNet for Modelling Multichannel Noise:** The UNet architecture (Ronneberger et al., 2015) is a well-established backbone in diffusion models due to its ability to maintain size consistency between noisy inputs and their corresponding denoised outputs. For our multichannel data, we extend the traditional 3DUNet (Özgün Çiçek et al., 2016) into a dual-channel 3D architecture, which we refer to as "DualChannelUNet." This adapted network is employed during the denoising diffusion process to jointly process the unquantised latent representations of the cell and nucleus channels. Specifically, the input to the DualChannelUNet consists of paired tensors representing the unquantised latent features of both the cell and nucleus channels. To effectively capture the 3D structure inherent to the data, we replace the original 2D convolutional layers of UNet with 3D convolutions, enhancing spatial and volumetric feature extraction across both channels simultaneously.

Additionally, drawing inspiration from Khader et al. (2023), we incorporate spatial- and depth-wise attention layers within the DualChannelUNet architecture. These attention layers are placed within the downsampling and upsampling paths, as well as the middle processing block, to capture both global and local dependencies between channels. By adaptively highlighting critical features, such as cell boundaries or nucleus structures, these mechanisms enhance the model's ability to generate biologically consistent and high-fidelity 3D volumes.

**Final assembly to "Build Your Own Cell" (BYOC):** In our BYOC pipeline, we propose a novel approach for multichannel 3D synthesis using independently learned codebooks for the cell and nucleus channels. These codebooks store channel-specific latent representations, encoding distinct features within each channel. The dependencies between the cell and nucleus are captured during the diffusion process. Before decoding, the unquantised latent features for each channel are passed through a 3D multichannel diffusion model, where the DualChannelUNet architecture processes the inputs. The DualChannelUNet, a dual-channel 3DUNet variant, ensures efficient spatial and volumetric feature extraction for both channels. Following the approach of Khader et al. (2023), the unquantised latents, represented as a paired tensor for the cell and nucleus, are normalised to a range of $-1$ to $1$ using the minimum and maximum values from their respective codebooks to stabilise the diffusion process. The reverse diffusion, starting from Gaussian noise, iteratively refines the latents, allowing the model to learn the interdependencies between channels. Finally, the refined latents are decoded into detailed 3D volumes, ensuring biologically accurate cell and nucleus reconstructions, thus completing the BYOC synthesis pipeline.

## 4 EXPERIMENTS

### 4.1 MATERIAL & IMPLEMENTATION

**Dataset:** Our dataset comprises over $7,083$ individual metastatic melanoma cells, imaged using light-sheet microscopy to capture detailed 3D reconstructions of both the cell body and nucleus. These single cells are extracted as cropped regions of interest from larger microscopy stacks, ensuring that the dataset focuses on individual cellular structures. The cropped cells are variable in size, reflecting the biological diversity and morphological heterogeneity present in the original stacks. The imaging resolution is $1\ \mu\mathrm{m}^3$, capturing fine cellular details and structures. The cells are embedded in tissue-like collagen matrices, providing a physiologically relevant environment that closely mimics natural tissue micro-environments. Each cell was treated with one of three different drugs - Nocodazole, Binimetinib, or Blebistatin- which induce distinct morphological changes. Cells treated with Nocodazole exhibit a round and flat structure, while those exposed to Blebbistatin develop a more spindly shape. The Binimetinib-treated cells present an intermediate morphology. This variation in drug response offers a rich dataset for studying the morphological effects of different treatments in a multichannel 3D context.

**Implementation Details:** The input volumes were padded to a size of $C \times 64 \times 64 \times 64$ for consistency. For each drug, the implementation of BYOC involved two distinct training phases. In the first phase, the VQGAN was trained end-to-end for $100,000$ timesteps with a batch size of 2, a learning rate of $3 \times 10^{-4}$, and a latent size of 16. After this phase, the weights of the encoder, codebooks, and decoder were frozen. For the second phase, we used a DualChannelUNet with a diffusion model (DDPM) configured for 1000 timesteps, trained with an $L_1$ loss, a learning rate of $1 \times 10^{-4}$, and a batch size of 2. The dataset was split into $80\%$ for training and $20\%$ for validation. All models were training using Pytorch Lightning on 4 nVidia Tesla V100 GPUs, each with 24GB of RAM.

### 4.2 QUALITATIVE EVALUATION

Evaluating the synthesis of biological samples lacks a widely accepted standard. To address this, we use both quantitative and qualitative evaluation methods. The qualitative evaluation compares our synthesised samples to those produced by the current state-of-the-art (SOTA) method in 3D medical image generation.

Depicted in Figure 4A, BYOC demonstrates superior performance in synthesising high-quality samples compared to the current state-of-the-art, MedicalDiffusion (Khader et al., 2023). While MedicalDiffusion captures the overall phenotypic structures, it struggles with the clarity of both the

nucleus and cell boundaries, which appear less distinct. In contrast, BYOC preserves these critical details more effectively, resulting in higher fidelity and sharper boundaries. The BYOC-generated samples exhibit strong morphological consistency, closely matching the phenotypic characteristics of the corresponding drug-treated cells. Although some finer details appear slightly smoother than in real samples, the generated samples maintain inter-channel consistency, with accurate positioning of the cell and nucleus. Additionally, BYOC effectively captures more complex structures, such as cells with elongated protrusions. Extending the qualitative evaluation, Figure 4B highlights the superior morphological accuracy, structural quality, and consistency achieved by the BYOC framework compared to MedicalDiffusion. Across all orthogonal views—axial, coronal, and sagittal—BYOC-generated samples outperform MedicalDiffusion, showcasing clearer structural details and more biologically realistic features.

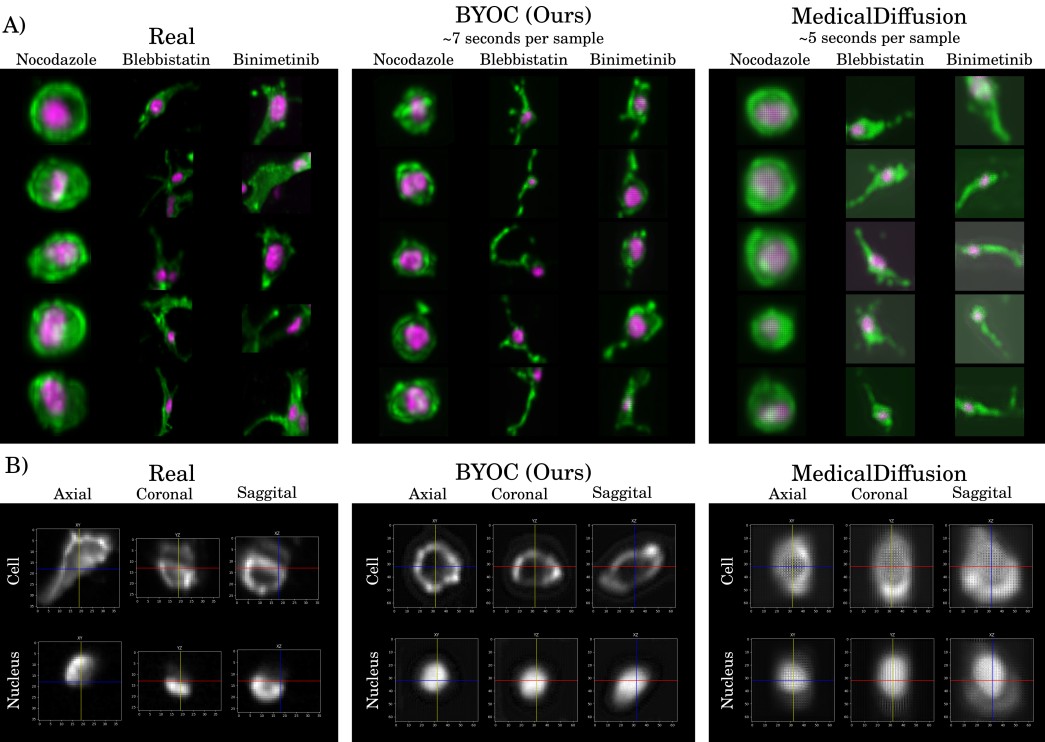

Figure 4: A) Qualitative comparison of our synthesised samples to the MedicalDiffusion (Khader et al., 2023) model, shown alongside ground-truth data. This illustrates the visual fidelity and accuracy of the generated samples relative to the actual biological structures of the different drug-treated melanoma cells. B) Orthogonal slices (axial, coronal, and sagittal) of 3D cell and nucleus volumes for real samples, BYOC-generated samples, and MedicalDiffusion-generated samples. The yellow, red, and blue lines represent the intersection of slices along the X, Y, and Z planes, respectively.

### 4.3 QUANTITATIVE EVALUATION

**Baselines:** In our quantitative evaluation, we compare the performance of BYOC against several baseline models, including HA-GAN (Sun et al., 2022), W-GAN (Arjovsky et al., 2017), $\alpha$-GAN (Gong et al., 2023), and MedicalDiffusion (Khader et al., 2023). These baselines were selected for their notable performance in synthesising biological data, as well as their diverse approaches to generative modelling. HA-GAN is a hierarchical adversarial network known for handling complex, structured data, while W-GAN and $\alpha$-GAN are widely adopted for their improvements in training stability and performance on high-dimensional data. MedicalDiffusion was included as it represents the most relevant comparison for 3D biological sample generation, specifically in the context of diffusion modelling. The inclusion of these models ensures a robust evaluation of BYOC across both adversarial and diffusion-based frameworks, providing a comprehensive benchmark against established methods.

| Nocodazole | Blebbistatin | Binimetinib |
|---|---|---|

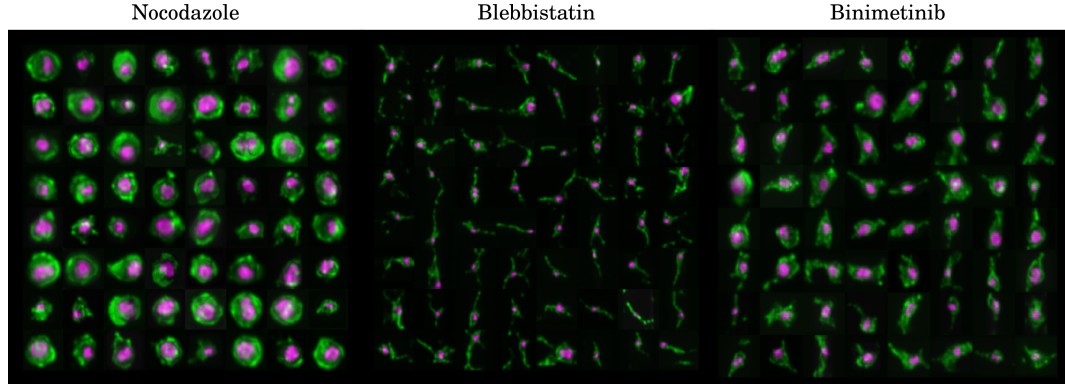

Figure 5: Samples of synthetic 3D cell structures generated by the BYOC framework for three different drug treatments: Nocodazole, Blebbistatin, and Binimetinib. The generated samples show distinct morphological characteristics specific to each drug, as well as sample diversity. The synthetic cells maintain clear nucleus positioning and boundary details, illustrating the effectiveness of the generative model in capturing 3D inter- and intra-channel biological structures.

**Metrics:** We test the quantitative realism of the generated samples using the Fréchet Inception Distance (FID) (Heusel et al., 2018) and Maximum Mean Discrepancy (MMD). FID quantifies the similarity between real and generated datasets by calculating the distance between their latent representations, which are extracted using Med3D (Chen et al., 2019), a pre-trained 3D medical imaging segmentation network trained on 8 different 3D medical segmentation datasets. Similarly, MMD measures the similarity of datasets by computing the distance between the means of their feature distributions.

For a consistent comparison, all generated samples were standardised to a size of $64^3$, requiring adjustments to the HA-GAN architecture to synthesise representations within this dimensional constraint. Additionally, both the synthetic and real samples were adjusted by taking a channel-wise average before calculating the FID and MMD metrics. For each method, 5000 samples were generated for evaluation.

Table 1: Quantitative comparison of different generative models across Nocodazole, Blebbistatin, and Binimetinib treatments, evaluated using 5-fold cross-validation. Results are shown in terms of Fréchet Inception Distance (FID) (Heusel et al., 2018) and Maximum Mean Discrepancy (MMD) ($\times 10^{-4}$). The best-performing scores, calculated as the average across folds, are shown in **bold**.

| Model | Nocodazole | | Blebbistatin | | Binimetinib | |
|---|---|---|---|---|---|---|
| | FID↓ | MMD ↓ | FID↓ | MMD ↓ | FID↓ | MMD ↓ |
| HA-GAN | $6.44_{0.05}$ | $30.54_{0.27}$ | $3.76_{0.05}$ | $16.11_{0.16}$ | $5.19_{0.05}$ | $23.97_{0.21}$ |
| W-GAN | $2.75_{0.04}$ | $12.74_{0.21}$ | $1.29_{0.03}$ | $3.96_{0.1}$ | $2.1_{0.03}$ | $9.00_{0.15}$ |
| $\alpha$-GAN | $2.73_{0.04}$ | $12.62_{0.21}$ | $1.3_{0.03}$ | $4.00_{0.1}$ | $2.1_{0.03}$ | $9.00_{0.15}$ |
| MedicalDiffusion | $2.12_{0.03}$ | $9.55_{0.17}$ | $2.26_{0.69}$ | $9.07_{3.5}$ | $1.62_{0.03}$ | $6.63_{0.13}$ |
| BYOC | $\mathbf{1.91}_{0.03}$ | $\mathbf{8.34}_{0.15}$ | $\mathbf{0.99}_{0.03}$ | $\mathbf{2.82}_{0.08}$ | $\mathbf{1.43}_{0.02}$ | $\mathbf{6.06}_{0.1}$ |

The quantitative results in Table 1 compare the performance of BYOC, HA-GAN, W-GAN, $\alpha$-GAN, and MedicalDiffusion across three drug treatments: Nocodazole, Blebbistatin, and Binimetinib, using FID and MMD. BYOC consistently achieves the best scores across both metrics, demonstrating its capacity to synthesise biologically realistic and diverse samples. This performance is attributed to the model's diffusion-based approach and its novel library of codebooks, which independently process each channel while simultaneously learning inter-channel dependencies. By comparison, GAN-based methods such as HA-GAN and $\alpha$-GAN struggle to maintain similar levels of coherence, while MedicalDiffusion, although effective, performs less consistently across all drug treatments. These results validate BYOC as a robust framework for generating high-quality multichannel 3D cellular data.

## 4.4 Ablation Study

To better understand the influence of the library of codebooks, we generated samples using different quantised representations. These quantised representations, derived directly from a library of learned codebooks, play a critical role in facilitating the latent diffusion denoising process during inference. To evaluate the impact of these codebooks on the quality of the generated samples, we systematically compared their performance across different metrics and drug treatments. "Unimodal" refers to the use of a single codebook that learns the average representations of both channels. "Absolute" involves separate codebooks for the cell and nucleus channels, with the representations normalised using the absolute values of both codebooks. The "Cell" and "Nucleus" implementations learn separate codebooks for each channel during training, but only a single codebook (either the cell or nucleus codebook) is used dur-

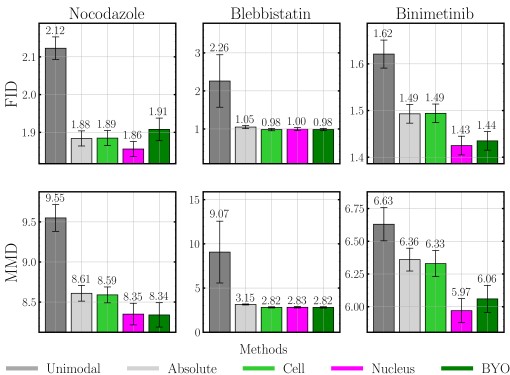

Figure 6: Comparison of performance metrics (FID and MMD) across different codebook implementations.

ing inference to normalise both channels. Finally, BYOC combines both cell and nucleus codebooks, capturing interdependencies between the two channels for improved representation and synthesis. Depicted in Figure 6, our investigation revealed that using a library of codebooks consistently outperforms the unimodal setting, where only a single codebook is used. Interestingly, in cases such as the Binimetinib treatment, the nucleus codebook alone demonstrated the ability to encode sufficient information to reconstruct the entire cell representation, highlighting the nucleus's central role in capturing morphological features under certain drug conditions. This ablation showcased that constructing codebooks that are specific to a biological channel enhances reconstruction over a "globally-represented' (unimodal) codebook. Furthermore, understanding the influence of an individual codebook (or combinations thereof) from a wider library of codebooks reveals subtle phenotypic characteristics that best represent a specific treatment.

## 5 Conclusion

This research introduced a robust generative framework, BYOC, specifically designed for the synthesis of biologically realistic multichannel 3D cell structures. By leveraging a unique combination of a "library of vector quantised codebooks" and "multichannel diffusion-based modelling," our approach significantly improved performance in terms of morphological consistency, structural realism, and fine-grained detail preservation, particularly across varied drug treatments. Compared to existing state-of-the-art methods, BYOC demonstrated its ability to capture complex phenotypic diversity, ensuring precise representation of critical features such as nucleus and cell boundary integrity. The resulting synthetic data holds potential as a valuable tool for downstream biological analysis, enhancing the ability to study cellular morphology and screen drug responses in silico.

**Limitations & Future Directions:** This study is presently limited to two channels, focusing on the cell body and nucleus, which constrains its applicability to more complex multichannel datasets encompassing additional organelles or cellular compartments. Extending the framework to accommodate more channels could enable broader biological insights. Furthermore, the evaluation was restricted to drug-treated melanoma cells, limiting its applicability to other cell types or treatment conditions. Future work should aim to evaluate this approach across diverse biological contexts and incorporate alternative diffusion models to improve scalability and performance. Additionally, generating samples derived from combinations of codebooks holds promise for exploring novel phenotypic states or treatment interactions. Another exciting avenue is adapting the framework for 4D data, enabling dynamic simulations of cellular behaviour over time. These enhancements could significantly advance applications in pre-clinical research, drug development, and personalised medicine.

### ACKNOWLEDGMENTS

Use unnumbered third level headings for the acknowledgments. All acknowledgements, including those to funding agencies, go at the end of the paper.

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

# A  APPENDIX

## A.1  TRAINING DETAILS

**BYOC Implementation:** All models were trained using mixed precision (fp16) with gradient checkpointing to manage memory usage efficiently. The dataset consisted of $7,083$ single-cell human melanoma samples (WM266.4), categorised by drug treatment: $2,314$ Nocodazole-treated cells, $2,264$ Blebbistatin-treated cells, and $2,504$ Binimetinib-treated cells. Identical train/val splits were employed across all baseline models to ensure consistency in performance evaluation. The hyperparameters for our model are detailed in Tables 2 and 3

**Basline Implementation:** Where applicable, baseline models were adjusted to process multichannel inputs. This primarily involved modifying the 3D convolutional layers of each architecture to accommodate two channels, representing both the cell and nucleus. These models were trained with the same data, ensuring a fair comparison across methods.

Table 2: VQGAN Hyperparameters

| Hyperparameter | Value |
|---|---|
| Learning Rate | $3 \times 10^{-4}$ |
| Batch Size | 2 |
| Latent Dimension (per channel) | 16 |
| Training Steps | 100,000 |
| Codebook Size (per codebook) | 1024 |
| Reconstruction Loss | Mean Squared Error (MSE) |
| Commitment Loss Weight | 0.25 |
| Optimizer | Adam |
| Beta 1 (Adam) | 0.9 |
| Beta 2 (Adam) | 0.99 |

Table 3: 3D DualChannelUNet Hyperparameters

| Hyperparameter | Value |
|---|---|
| Learning Rate | $1 \times 10^{-4}$ |
| Batch Size | 2 |
| Number of Timesteps | 1000 |
| Loss Function | L1 Loss |
| Number of Channels | 2 (Cell, Nucleus) |
| 3D Convolution Kernel Size | $3 \times 3 \times 3$ |
| Dimension Multiplier | [1,2,4,8] |
| Number of Attention Layers | 2 (Spatial and Depth-wise) |
| Optimizer | Adam |
| Beta 1 (Adam) | 0.9 |
| Beta 2 (Adam) | 0.99 |
| Normalization | Instance Normalisation |
| ema decay | 0.995 |

## A.2  EVALUATION DETAILS

The evaluation of our generative model's ability to synthesise realistic 3D cell structures involves a rigorous quantitative assessment using established metrics. Specifically, we calculate two key metrics to evaluate the quality of the synthetic 3D cellular structures:

1. Fréchet Inception Distance (FID): This metric measures the similarity between the distributions of real and generated samples by comparing their feature representations. FID is widely used in generative modelling, particularly in image synthesis tasks, where lower FID values indicate a closer resemblance between the generated and real samples.

2. Maximum Mean Discrepancy (MMD): This kernel-based method compares the similarity between two distributions—in this case, the real and synthetic data. Lower MMD values indicate higher similarity between the distributions, providing an additional quantitative measure of quality.

To compute these metrics, we extract feature representations of the real and synthetic 3D volumes using the Med3D framework (Chen et al., 2019). Med3D is a pre-trained ResNet50 model specifically designed for 3D medical imaging tasks and trained on eight diverse 3D segmentation datasets. It is widely employed for feature extraction in this domain (Tudosiu et al., 2024) due to its ability to capture high-dimensional representations of 3D structures across multiple layers. For each 3D volume, the Med3D model processes the input, and its feature maps are spatially averaged across the height, width, and depth dimensions to generate a compact feature vector that represents the 3D structure. These feature vectors are then concatenated into a single tensor for subsequent metric calculations. This approach ensures that the metrics effectively capture the morphological and structural nuances of the synthetic 3D cellular structures.

## A.3 SYNTHESISED EXAMPLES

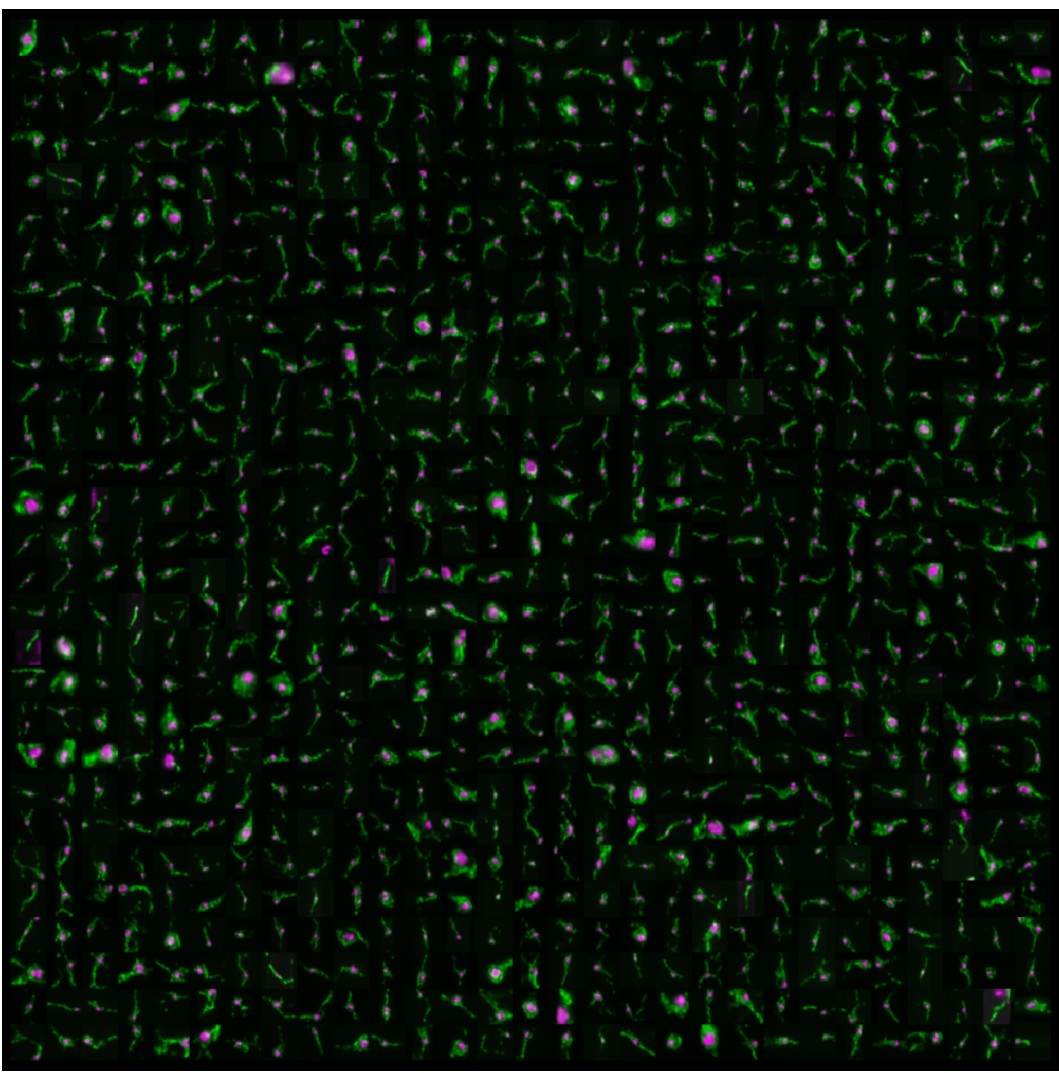

Figure 7: A library of synthesised examples produced from BYOC.

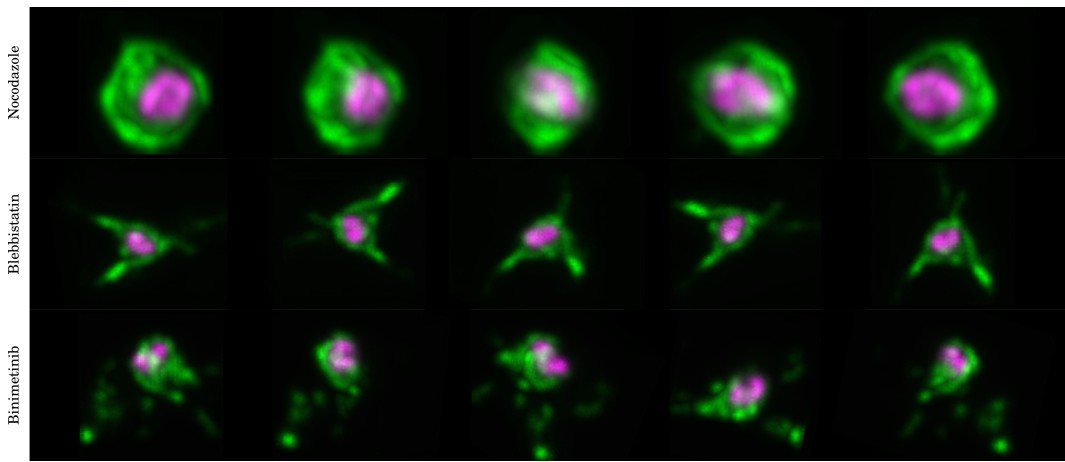

Figure 8: BYOC-generated samples of each drug from different views.

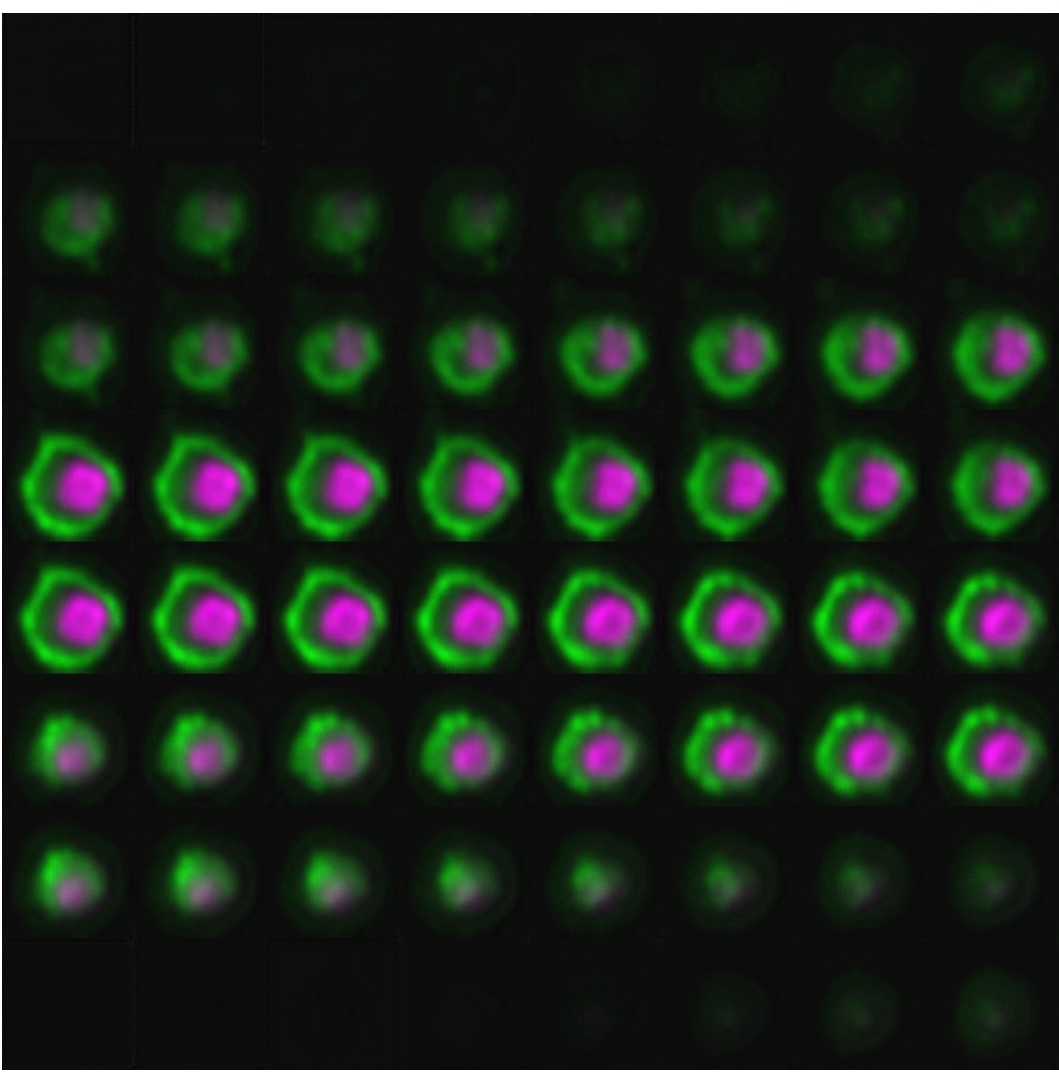

Figure 9: Multichannel visualisation of BYOC-generated 3D cell and nucleus structure across 64 depth planes for Nocodazole.

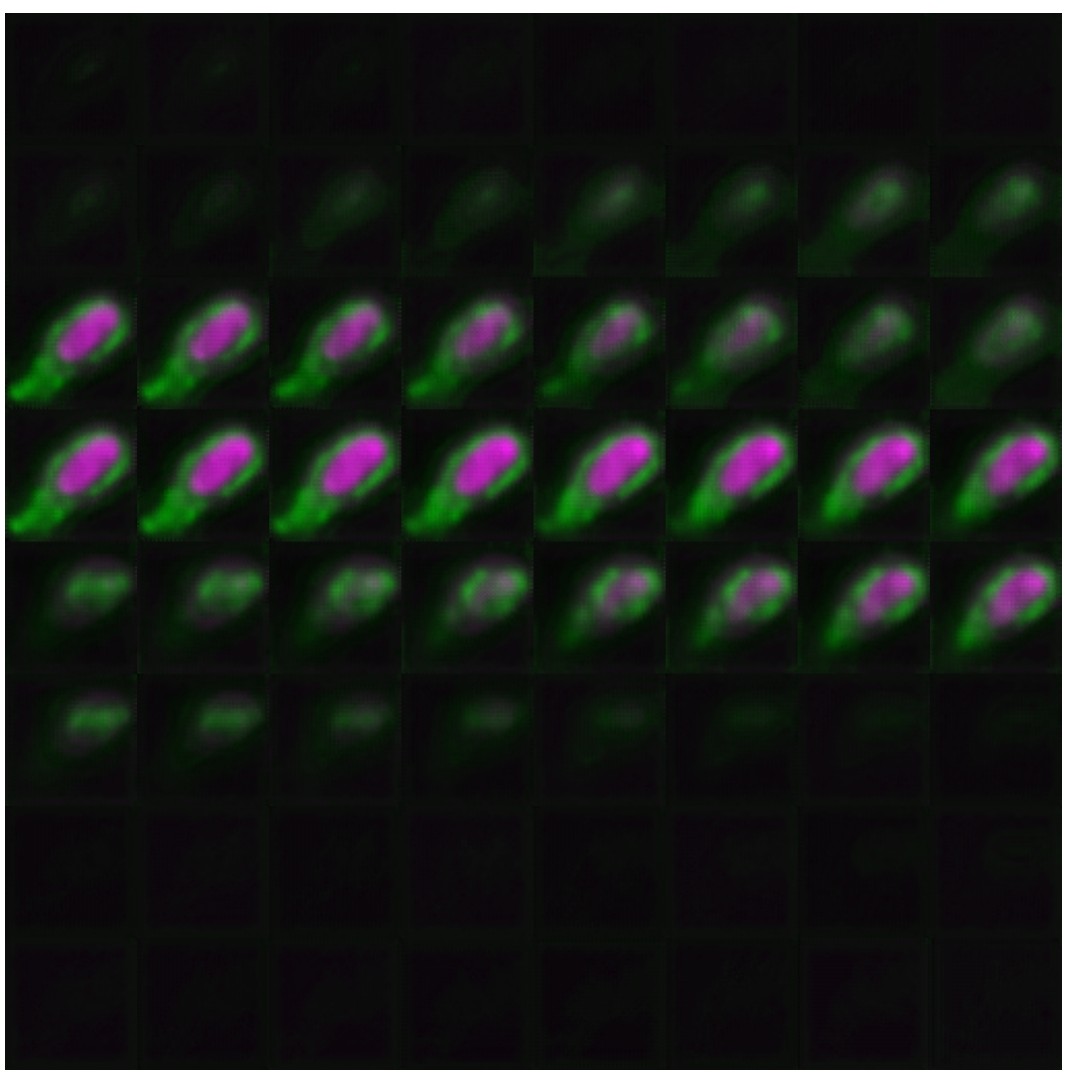

Figure 10: Multichannel visualisation of BYOC-generated 3D cell and nucleus structure across 64 depth planes for Binimetinib.

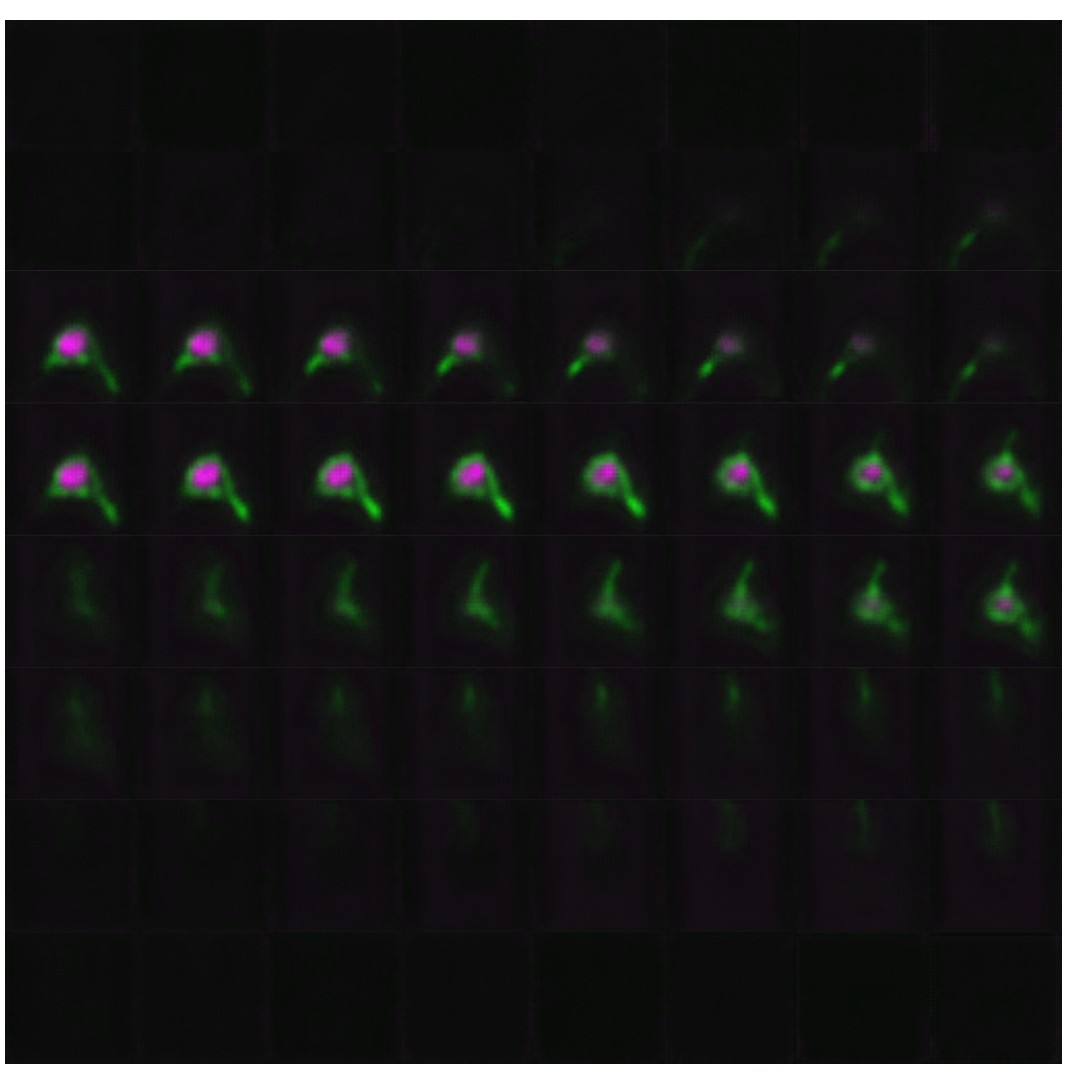

Figure 11: Multichannel visualisation of BYOC-generated 3D cell and nucleus structure across 64 depth planes for Blebbistatin.

