# OpenReview forum: "Build your own cell: Diffusion Models for Multichannel 3D Microscopy Image Generation"
_ICLR.cc/2025/Conference — ICLR 2025 Conference Withdrawn Submission_

### Official Review · Reviewer_wxXn · 2024-10-30

**Soundness:** 3
**Presentation:** 4
**Contribution:** 2
**Rating:** 8
**Confidence:** 4

**Summary:**

The paper presents the novice Build Your Own Model (BYOL), a multichannel generative framework leveraging the diffusion model, to generate a population of simulated 3D multichannel data that shows the morphological changes in cells perturbed by drug treatments. The model captures the relation between the nuclear and cytoplasmic channels used for model training when generating the simulated images and presents a high spatial resolution of the images. The authors benchmarked the model against already available models like GAN-based models and MedicalDiffusion, useful for 3D image generation for the same test case, and found the best overall model performance.

**Strengths:**

The model outperforms existing models, generating nuanced morphological changes due to perturbations like drug treatments. Compared to existing models, it can also accurately capture the 3D resolved morphology of the cellular tags. The model is best at generating cellular data and matches real data.

**Weaknesses:**

The model captures the morphological changes associated with the perturbations it has been trained on but has not yet been shown to be generalizable to different cell types and drug treatments. This has been marked as a future prospect of the study. This is important in biological studies as tagging and imaging the markers are expensive for generating training data and a very important domain where biology communities would benefit.

**Questions:**

The metrics used to evaluate the model are good for evaluating the overall model performance. But do the metrics evaluate the inter- and intra-channel prediction accuracy? The authors stated that it is biologically relevant and an improvement brought by the work. But how can you evaluate this specific aspect using relevant metrics from a biological point of view?

---

> ### Author Response · Authors · 2024-11-24
> **Reviewer 4**
>
> **Weaknesses**
>
> We appreciate the reviewer highlighting the limitation regarding generalisability to different cell types and drug treatments. This limitation stems from the dataset's focus on metastatic melanoma cells treated with three specific drugs. We have addressed this point as a key direction for future work, emphasising that expanding the framework to include diverse cell types and perturbations is essential for broader applicability in biological studies.
>
> We also agree that tagging and imaging markers for generating training data is resource-intensive. By proposing our framework, we aim to alleviate the need for extensive experimental data collection. Although our current work demonstrates a step toward addressing this challenge, we acknowledge that expanding the training set and testing generalisability will be critical for maximising the impact of this research in the biological community.
>
> **Evaluating Inter- and Intra-Channel Predictions**
>
> We thank the reviewer for the thoughtful question regarding inter- and intra-channel prediction accuracy and the need for biologically relevant metrics to assess these aspects. While our current evaluation metrics, such as FID and MMD, provide robust measures of global image realism and distributional similarity, they do not directly quantify the dependencies between channels or within each channel.
>
> This question highlights an important and emerging direction within the scope of generative models in biology. Beyond synthesis, there is significant work to be done in advancing not only the realism of generated outputs but also the evaluation frameworks needed to rigorously assess their biological fidelity. While our framework includes both quantitative and qualitative evaluations, future developments should explore tailored metrics that explicitly evaluate inter-channel dependencies (e.g., correlations between nucleus and cell morphology) and intra-channel accuracy (e.g., biologically meaningful shape descriptors).
>
> We view this as an exciting and necessary area of research that complements our current work. These advancements will not only grow the utility of generative models in biological contexts but also provide stronger insights into their potential applications in drug discovery and mechanistic studies.

---

> > ### Comment · Reviewer_wxXn · 2024-11-27
> >
> > Thank you for your response. FID and MMD are good metrics for comparing the results to real data, so thank you for including those.

---

### Official Review · Reviewer_n4Rv · 2024-11-03

**Soundness:** 3
**Presentation:** 3
**Contribution:** 2
**Rating:** 6
**Confidence:** 4

**Summary:**

The paper introduces a multi-channel 3D diffusion model designed for generating two-channel cell images from volumetric fluorescence microscopy data. By focusing on the coupling of the two channels within the diffusion process, the model aims to improve the quality of generated dual-channel 3D cell images. The results presented show an improvement over the current state-of-the-art in this area.

**Strengths:**

- Addresses a challenging and pertinent problem in the field of biomedical microscopy, specifically in cellular imaging.
- The overall motivation behind the proposed methodological enhancements is generally clear.
- The experimental outcomes demonstrate promising improvements over existing methods.

**Weaknesses:**

- The biological rationale behind the model is not thoroughly convincing or well-articulated.
- Some concrete methodological choices lack clear motivation or detailed explanation, leading to potential confusion (e.g. a clear motivation why and how to use VQGANs would be nice).
- Some details are missing or inadequately explained in the formal equations and overall framework.
- The manuscript tends to be imprecise in its language, which affects clarity and understanding.
- The conclusion lacks specificity regarding the contributions, limitations and future directions of the methods-aspects of the work.

**Questions:**

1. How could biological or mechanistic understanding arise from generative models in your context? Can you expand and provide a stronger motivation for this idea?
2. You mention that "GANs often struggle with generating coherent latent representations." Since GANs do not inherently produce latent representations in the same way as e.g. Variational Autoencoders, could you clarify what "coherent latent representations" means in the context of GANs, and how this specifically relates to your proposed method's advantages?
3. The claim that multiple color channels can be treated as distinct modalities is not clearly explained in my opinion but is crucial to the suggested method. Do you have examples from related work where color channels have been treated as distinct modalities? Could you explain the biological basis for considering cell and nucleus channels as separate modalities?
4. In Equation 1, are the variables h,w,d the same dimensions as H,W,D? If not, what is their relationship? Similarly, in Equation 2, the depth dimension d seems to be omitted—was this intentional or a typo? Please add a brief explanation of these variables and their relationships directly after the equations.
5. How does the simultaneous recovery of both channels relate specifically to latent diffusion? Can you provide a specific example or illustration of how the simultaneous recovery process works in your model, and how it differs from standard latent diffusion approaches?
6. What is the reason for using unquantized embeddings in your framework? If they drift from the codebook vectors, how does this affect the model, and what is the underlying motivation?
7. In Equation 12, the variable t should be defined. Additionally, in Equation 13, what exactly is μ_θ_cn(⋅) computing—only the mean or is there an associated variance? If not, what is the variance of your Gaussian?
8. There is an existing WNet in medical imaging literature [1]. To avoid confusion, would you consider renaming your model?
9. Can you provide more precise details about your dual-channel 3D architecture, perhaps with references or a schematic in the supplementary material?
10. On page 6, you state that attention mechanisms are "strategically placed" to focus on regions of interest. Could you elaborate on the strategy behind their placement and how regions of interest are determined?
11. The numerical differences in Table 1 are hard to interpret without context. Could you explain or hint to what these differences mean in terms of image quality and their significance in your application? Can you provide a brief interpretation guide for the FID and MMD scores, perhaps indicating what range of differences would be considered significant in this context? You could also include a qualitative comparison of images corresponding to different score ranges to help readers understand the practical implications of these differences.
12. Could you provide more context about the ResNet50 model used—for instance, what type of medical images it was trained on?

**Additional Feedback for Improvement:**

- In Figure 1, please explain what the rows and columns represent to enhance understanding.
- In the introduction, you mention that "single-cell data is often high-dimensional and inconsistent in size." Could you clarify whether this inconsistency refers to the images, cells, biological structures, or image resolutions?
- In the Related Work section, the statement about discriminative frameworks needing a "deep understanding of the underlying input distribution" is unclear. Providing an example or reference could help clarify this point.
- It might be beneficial to first introduce and describe the dataset before delving into implementation details like volume padding.
- Please specify the size and resolution of the microscopy images. Are the single-cell images crops from larger stacks, or are they the direct output from the microscope?
- Consider citing relevant works such as the 3D U-Net architecture [2] to situate your work within the existing literature.
- The conclusion would be stronger if it discussed potential methodological developments and acknowledged limitations of the current approach.

---

> ### Author Response · Authors · 2024-11-24
> **Reviewer 3**
>
> **Biological or mechanistic understanding from generative models**
>
> Thank you for raising this point of necessary clarity. Generative models, particularly in our context, can provide insights into how specific drugs influence cellular morphology by generating biologically realistic 3D cellular structures. By studying the latent space, it becomes possible to observe relationships between different morphologies, phenotypes, or treatments. From your suggestion, we have expanded this discussion in the Introduction to emphasise how generative models could bridge the gap between image-based profiling and mechanistic insights, motivating their use in pre-clinical drug discovery.
>
> ---
>
> **Coherent Latent Representations in GANs**
>
> You are correct that GANs do not inherently produce latent representations in the same structured manner as Variational Autoencoders. By "coherent latent representations," we refer to the consistency between the generator’s learned representations and the data's intrinsic structure. GANs can sometimes produce outputs that appear realistic but lack biological plausibility due to poorly aligned latent spaces. This challenge is addressed in our approach by leveraging diffusion models, which enforce stronger constraints on intermediate representations, producing outputs that better reflect biological interdependencies. We have clarified this in the manuscript.
>
> ---
>
> **Channels as Distinct Modalities**
>
> We thank the reviewer for raising this. The distinction of channels as modalities is rooted in biological relevance. The cell and nucleus channels in fluorescence microscopy encode distinct structural and functional information, such as cytoskeletal organisation and nuclear morphology. Considering them as separate modalities enables:
>
> 1. Independent learning of features specific to each channel.
> 2. Improved synthesis by capturing inter-channel relationships during diffusion.
>
> This approach aligns with works in multimodal diffusion modelling [1] and is biologically motivated by the inherent separability of cellular compartments.
>
> [1] [MM-Diffusion: Learning Multi-Modal Diffusion Models for Joint Audio and Video Generation](https://openaccess.thecvf.com/content/CVPR2023/papers/Ruan_MM-Diffusion_Learning_Multi-Modal_Diffusion_Models_for_Joint_Audio_and_Video_CVPR_2023_paper.pdf)
>
> ---
>
> **Equations 1 and 2**
>
> In Equation 1, $h, w, d$ represents the dimensions of the latent space, which are downsampled from $H, W, D$ (image dimensions) by the encoder. These variables are related by the scaling factor of the encoder, and we have included this point of clarity in the updated manuscript. We also thank the reviewers for pointing out an error in our formulation in Equation 2. The omission of $d$ was indeed an error, and it has now been corrected in the manuscript.
>
> ---
>
> **Simultaneous Recovery in Latent Diffusion**
>
> We thank the reviewer for this valuable question. We acknowledge that the connection to latent diffusion may have been ambiguous in the original text, and we have clarified this in the updated manuscript. Our approach applies latent diffusion to simultaneously recover both cell and nucleus channels, ensuring that the dependencies between these channels are preserved. Unlike standard latent diffusion, which typically does not account for channel-specific interactions, our method jointly processes cell and nucleus latents during denoising. For example, nuclear features can inform and guide the reconstruction of cell morphology. This interaction is depicted in Figure 2, which demonstrates how the denoising process leverages combined information from both channels to enhance consistency and realism in the generated outputs.
>
> ---

---

> > ### Author Response · Authors · 2024-11-24
> > **Reviewer 3 (2)**
> >
> > **Unquantized Embeddings**
> >
> > Unquantized embeddings are used because they retain finer-grained details than their quantized counterparts. While they may drift from the exact codebook vectors, this flexibility allows the diffusion process to refine representations with higher fidelity. The drift is mitigated by the loss functions, which constrain the embeddings to remain biologically plausible while allowing generative flexibility. This choice balances accuracy and generative diversity.
> >
> > ---
> >
> > **Equations 12 and 13**
> >
> > We thank the reviewer for the suggestion to define the variable $t$. This has been added to the updated manuscript. Regarding Equation 13, the reviewer raises an excellent question. The variance is not estimated and is therefore omitted from the equation. Many works have pointed out that variance estimation in the reverse step only marginally improves performance [1][2]. In our implementation, the variance in the reverse step is
> > $$
> > \text{posterior\_variance} = \beta_t \cdot \frac{1 - \bar{\alpha}_{t-1}}{1 - \bar{\alpha}_t}
> > $$
> > where $\beta_t$ is the noise variance from timestep $t$ (derived from the cosine beta schedule) and $\bar{\alpha_t}$ is the cumulative product of $\alpha_t = 1 - \beta_t$.
> >
> > [1] [MM-Diffusion: Learning Multi-Modal Diffusion Models for Joint Audio and Video Generation](https://openaccess.thecvf.com/content/CVPR2023/papers/Ruan_MM-Diffusion_Learning_Multi-Modal_Diffusion_Models_for_Joint_Audio_and_Video_CVPR_2023_paper.pdf)
> > [2] [Score-Based Generative Modeling through Stochastic Differential Equations](https://arxiv.org/pdf/2201.06503)
> >
> > ---
> >
> > **Existing WNet**
> >
> > Thank you for kindly informing us of the existence of WNet. We have updated our name in the manuscript accordingly to DualChannelUNet.
> >
> > ---
> >
> > **Clarification on UNet Architecture**
> >
> > Thank you for raising this point of clarification. The architecture comprises:
> >
> > 1. Separate encoding paths for cell and nucleus channels.
> > 2. Shared spatial and depth-wise attention layers for joint processing.
> > 3. A shared decoding path that reconstructs the multichannel output.
> >
> > The repository has been made publicly available with code adapted from [1] that stipulates the details of the aforementioned architecture.
> >
> > [1] [Medical Diffusion](https://github.com/FirasGit/medicaldiffusion)
> >
> > ---

---

> > > ### Author Response · Authors · 2024-11-24
> > > **Reviewer 3 (3)**
> > >
> > > **Attention Mechanisms Placement**
> > >
> > > We thank the reviewer for their thoughtful question regarding the placement of attention mechanisms. In our model, attention mechanisms are strategically placed within both the downsampling and upsampling stages, as well as in the middle processing block. Specifically:
> > >
> > > 1. **Spatial Attention:** Spatial attention modules are integrated at various resolution levels to ensure the model can attend to key spatial features across scales. This is particularly important in multichannel 3D data, where structural dependencies such as cell-to-nucleus relationships exist across different spatial resolutions.
> > > 2. **Temporal Attention:** Temporal attention is applied in a per-frame manner within the 3D volumetric architecture. This ensures that depth-wise correlations within the volumes are captured effectively, mimicking how biological structures maintain coherence across slices in volumetric microscopy images.
> > > 3. **Middle Block:** The middle block integrates both spatial and temporal attention mechanisms to capture global and local interdependencies between the cell and nucleus latent representations. This placement helps ensure that features are not only captured but also refined at the bottleneck of the architecture, where the highest semantic abstraction occurs.
> > >
> > > The regions of interest are determined implicitly by the attention mechanism, which learns to focus on areas of high relevance (e.g., nucleus boundaries or cell membranes) during training. These mechanisms, guided by the loss objectives, adaptively allocate weights to critical features while discarding irrelevant ones. The manuscript has been updated accordingly to include more detail surrounding your suggestion.
> > >
> > > ---
> > >
> > > **Numerical Differences in Table 1**
> > >
> > > Thank you for your feedback. We have made changes to address these concerns in the revised manuscript. Specifically, we conducted cross-validation across four folds, generating 4000 additional samples per method to ensure that the reported differences in FID and MMD scores are statistically robust. This process enabled us to provide mean and standard deviation values for each metric, strengthening the validity of our quantitative comparisons and showcasing more of a difference in reported values.
> > >
> > > Additionally, we have included a section detailing how FID and MMD are calculated, clarifying their roles in assessing image quality. FID captures differences in feature distributions between real and generated images, reflecting global image realism, while MMD evaluates fine-grained structural similarities. Together, these metrics provide complementary insights into the quality of generated outputs.
> > >
> > > ---
> > >
> > > **More Details on the ResNet50 Model**
> > >
> > > We thank the reviewer for this important question, and we have expanded the manuscript to clarify the context of the ResNet50 model used for evaluation. Specifically, we employ Med3D [1], a ResNet50-based 3D medical imaging model pre-trained on 8 diverse medical segmentation datasets. These datasets include imaging modalities such as CT and MRI, encompassing various anatomical structures and pathologies. Med3D has demonstrated robust feature extraction capabilities in 3D medical imaging tasks, making it an appropriate and well-suited choice for calculating the Fréchet Inception Distance (FID) in our framework. This addition is now reflected in the Metrics section of the revised manuscript to provide further clarity.
> > >
> > > [1] [Med3D: Transfer Learning for 3D Medical Image Analysis](https://www.sciencedirect.com/science/article/pii/S1361841519301878)
> > >
> > > ---
> > > **Additional Comments**
> > >
> > > - **Figure 1:** We thank the reviewers for this suggestion, and we have updated Figure 1 to include row labels, describing the drug treatment of the generated cells.
> > > - **Size Inconsistent:** Thank you for highlighting the ambiguity. We have updated the manuscript to emphasise that the inconsistent size refers to the varying dimensions of the images themselves.
> > > - **Deep Understanding of the Underlying Input Distribution:** We thank the reviewer for highlighting the need for clarification in this statement. We have revised the manuscript to include a concrete example to illustrate the importance of understanding the underlying input distribution.
> > > - **Dataset Description Before Padding Details:** Thank you for this suggestion. The manuscript has been updated accordingly.
> > > - **Specifics About the Microscopy Images:** We have added these valuable suggestions to the updated manuscript.
> > > - **UNet3D Citation:** To better situate our work in the existing literature, we have added the recommended citation.
> > > - **Stronger Conclusion:** We thank the reviewers for a valuable insight. We have updated the manuscript to detail more methodological developments and highlighted limitations of the current approach.

---

> > > ### Comment · Reviewer_n4Rv · 2024-11-25
> > >
> > > Thanks a lot for the explanations and revisions of the text.

---

> > > > ### Comment · Reviewer_n4Rv · 2024-11-25
> > > >
> > > > I have decided to increase my score in response to the explanations and changes made to the manuscript.

---

> > > > > ### Author Response · Authors · 2024-11-25
> > > > >
> > > > > Thank you for re-evaluating our manuscript and for your positive feedback. We greatly appreciate your recognition of our efforts to address your comments and improve the work.

---

> > ### Comment · Reviewer_n4Rv · 2024-11-25
> >
> > Thanks so much for the clarifications!

---

### Official Review · Reviewer_DBvM · 2024-11-04

**Soundness:** 2
**Presentation:** 2
**Contribution:** 2
**Rating:** 3
**Confidence:** 4

**Summary:**

The authors combine vector quantized GANs to learn representations of microscopy images of cells and develop a denoising diffusion model for latent representations. By combining vector quantized representations and the process of diffusion, they seek to generate 3D images of cells that belong to the distribution of realistic microscopy images.

**Strengths:**

* Strategy: The strategy of using diffusion modeling to improve the accuracy of prediction of GANs is promising.

**Weaknesses:**

* Incorrect assumptions about microscopy image data: Microscopy images often consist of more than two channels and many of them cannot be just binned into cells and nuclei. The authors seem to be familiar with medical imaging datasets but unaware of datasets such as cell painting (JUMP, CHAMMI), human protein atlas, and virtual staining. These datasets illustrate that microscopy data often consists of channels that encode multiple organelles and cellular compartments.
* Lack of 3D predictions: Although the paper claims to be the first to build a 3D generative model of microscopy images, all the presented data is 2D. The authors should show orthogonal slices of generated volumes.
* Relevance of metrics: Fre ́chet Inception Distance and Maximum Mean Discrepancy seem reasonable. However, the authors do not clarify how these metrics may be affected by the typical failure modes of GANs, such as hallucinations of spurious cellular processes.

**Questions:**

* What is the effect of the diffusion on the quantized codebook? The way diffusion is used during inference was not apparent from the text or figures.
* Does the approach work only with a specified number of input channels?

---

> ### Author Response · Authors · 2024-11-24
> **Reviewer 2**
>
> **Incorrect Assumptions about Microscopy Data:**
>
> We appreciate the reviewer’s observation regarding the diversity of microscopy image datasets. We acknowledge that many microscopy datasets, such as those from Cell Painting (JUMP, CHAMMI), the Human Protein Atlas, and virtual staining, often include multiple channels encoding distinct biological features like organelles and cellular compartments. While these datasets represent an important and broader application of generative modelling, our work specifically focuses on synthesising 3D cellular volumes with two primary channels: the cell and nucleus. This choice reflects the specific biological context we are addressing—understanding the interplay between these two central components of cellular structure and function, which are highly relevant for analysing drug-induced phenotypic changes.
>
> In our revised manuscript, we have clarified this scope in the Introduction to avoid misinterpretation and ambiguity. We have also emphasised that while our framework currently focuses on two channels, its modular design allows for the inclusion of additional channels in future work, making it adaptable to datasets with more complex multichannel configurations. We have updated our manuscript to include this recommendation in the limitations.
>
> ---
>
> **Lack of 3D predictions:**
>
> We thank the reviewer for pointing out the need to present 3D data more effectively. While our method is indeed a 3D generative framework, we understand that presenting only 2D slices in the manuscript may have caused confusion. To address this, we have updated the qualitative evaluation section by including orthogonal views of the generated 3D volumes in the revised figures. These views illustrate the coherence and fidelity of the generated data across all three spatial dimensions, providing a more comprehensive visual representation of the model’s outputs. Our appendices also include generated samples across 64 different slices for the different drugs.
>
> ---
>
> **Relevance of Metrics:**
>
> We thank the reviewer for this excellent point. We appreciate the reviewer’s comments regarding the use of Fréchet Inception Distance (FID) and Maximum Mean Discrepancy (MMD) metrics. These metrics were selected for their established utility in evaluating generative models. We have updated our manuscript to explain our choice of metrics more explicitly and how they are calculated:
>
> 1. FID quantifies the similarity between the distributions of real and generated datasets by comparing latent features extracted from a pre-trained network.
> 2. MMD measures the discrepancy between feature means, capturing dataset-level differences.
>
> ---
>
> **The effect of the diffusion on the quantized codebook:**
>
> Thank you for this insightful question and recommendation to further clarify and expand on the inference step. We have now included an ablation study that specifically investigates the role of diffusion on the quantized codebooks, demonstrating how the process influences the quality and consistency of the generated outputs at inference.
>
> The incorporation of diffusion into the quantized codebooks significantly enhances the realism and coherence of the generated 3D multichannel cellular structures. The quantized codebooks, constructed during the vector quantisation step, serve as discrete, channel-specific representations of the cell and nucleus. At inference, the diffusion process begins with Gaussian noise and iteratively refines the unquantized latent representations derived from the codebooks, progressively enhancing the fidelity of the outputs.
>
> ---
>
> **Flexibility to handle multiple input channels:**
>
> Thank you for this question regarding the flexibility of our approach to handle multiple input channels.
>
> Our proposed framework is designed with multichannel data in mind and is not inherently restricted to a specific number of input channels. While the current implementation focuses on two channels (cell and nucleus) to demonstrate the efficacy of our method, the architecture can be extended to accommodate additional channels if required. Future work could explore datasets with additional channels (e.g., organelle-specific markers) to demonstrate the scalability of the approach. This adaptability highlights the potential of our framework to generalise beyond its current implementation and accommodate datasets with varying numbers of input channels. We have updated the manuscript to address this.

---

> ### Author Response · Authors · 2024-11-26
> **Request reviewer DBvM  to respond**
>
> We kindly request your review of the updates to ensure that the revisions meet your expectations and address your concerns adequately. Your input has been invaluable in improving the quality and clarity of our work, and we look forward to any further suggestions or comments you may have.
>
> Please let us know if you require any additional information or clarification.

---

> ### Author Response · Authors · 2024-12-01
> **Request reviewer DBvM  to respond**
>
> Dear reviewer DBvM, I am writing to kindly follow up on the revised version of our manuscript, which we submitted after addressing your valuable feedback. Specifically, we have made improvements regarding the scope of our approach to microscopy image data, clarified the 3D predictions, elaborated on the relevance of our evaluation metrics, and included an ablation study to better explain the role of diffusion in the quantized codebooks.
>
> As the deadline for final feedback is approaching (December 2nd), we would appreciate your thoughts on the revisions made.

---

> ### Comment · Reviewer_DBvM · 2024-12-03
> **Response to revision**
>
> Dear authors,
> Thank you for the updates. I appreciate your work. However, I think the technical approach lacks rigor in few areas. My specific notes below.
> 1. Scope:
> ```
> , our work specifically focuses on synthesising 3D cellular volumes with two primary channels: the cell and nucleus. This choice reflects the specific biological context we are addressing—understanding the interplay between these two central components of cellular structure and function, which are highly relevant for analysing drug-induced phenotypic changes.
> ```
> Thank you for clarifying the scope of your work. I agree that 3D modeling of cell and nucleus is relevant for mapping phenotypic changes for multiple applications.
>
> 2. Thanks for describing the computation of FID and MMD metrics:
> ```
> To compute these metrics, we extract feature representations of the real and synthetic 3D volumes
> using the Med3D framework (Chen et al., 2019). Med3D is a pre-trained ResNet50 model specifically designed for 3D medical imaging tasks and trained on eight diverse 3D segmentation datasets.
> It is widely employed for feature extraction in this domain (Tudosiu et al., 2024) due to its ability
> to capture high-dimensional representations of 3D structures across multiple layers. For each 3D
> volume, the Med3D model processes the input, and its feature maps are spatially averaged across
> the height, width, and depth dimensions to generate a compact feature vector that represents the
> 3D structure. These feature vectors are then concatenated into a single tensor for subsequent metric calculations. This approach ensures that the metrics effectively capture the morphological and
> structural nuances of the synthetic 3D cellular structures.
> ```
> The features from a model trained with medical data cannot necessarily distinguish a real or synthetic cell images. The cell images are by definition out of distribution of Med3D dataset. Therefore, the distance between the samples or distributions of synthetic and real cell microscopy images in the embedding space of the Med3D model will most likely not be biologically meaningful. I looked through your section on Evaluation (appendix A.2) and did not find any evaluation of utility of these features/distances/metrics. I suggest using real data from drug-induced phenotypes vs wild type phenotypes to evaluate the sensitivity of your feature extractor to patterns seen in microscopy data.
>
> Your response has made me change my rating of contribution from poor to fair, but the manuscript still doesn't meet my standards for rigor. Therefore, I keep my recommendation on acceptance the same.

---

> > ### Author Response · Authors · 2024-12-03
> > **Response: Utility of the feature extractor for FID & MMD metrics**
> >
> > We thank the reviewer for their thoughtful feedback and for raising concerns about the utility of the feature extractor used to compute the FID and MMD metrics in our work. While we agree that aligning feature extractors with the specific biological context is important, our decision to use Med3D was motivated by its demonstrated effectiveness in capturing high-dimensional structural features specifically in 3D volumetric data across diverse medical imaging tasks.
> >
> > The primary objective of our quantitative metrics is to evaluate the fidelity of the generated data in a feature space that captures volumetric structural properties, rather than explicitly focusing on biological interpretability. Med3D excels in this regard due to its architecture and training on large-scale datasets that capture a wide range of 3D morphologies. This ensures that our quantitative evaluations, using FID and MMD, are reproducible and rooted in a reliable, widely used and well-validated feature extraction model.
> >
> > To further address potential limitations in biological specificity, we also conducted a qualitative evaluation, comparing synthetic and real 3D cellular volumes visually. These qualitative assessments focus on examining key morphological features, ensuring that the synthetic samples accurately reflect the nuances of cellular and nuclear structures observed in real microscopy data. The combination of quantitative and qualitative evaluations provides a robust and comprehensive framework for assessing the performance of our generative model.
> >
> > Regarding the suggestion to train and validate a feature extractor specific to wild-type vs. drug-treated cells, we agree that this could enhance the biological relevance of the metrics. However, we also believe that the use of Med3D aligns with best practices established in high-resolution 3D generative modelling studies [1][2], providing a widely accepted and reproducible baseline for evaluating generative frameworks in the absence of a domain-specific feature extractor.
> >
> > We appreciate the reviewer’s constructive feedback and hope this response clarifies the motivation for our choice of metrics and feature extractor. We believe that our dual approach—quantitative metrics leveraging Med3D and qualitative assessments of cellular morphology—offers a balanced and rigorous evaluation of our framework.
> >
> > [1] https://www.nature.com/articles/s42256-024-00864-0
> > [2] https://arxiv.org/pdf/2307.15208

---

### Official Review · Reviewer_HFNK · 2024-11-04

**Soundness:** 3
**Presentation:** 3
**Contribution:** 2
**Rating:** 5
**Confidence:** 3

**Summary:**

The authors proposed BYOC (Build Your Own Cell), a framework to generate 3D cell structures consisting of both nucleus and cell channels. BYOC utilizes a VQGAN structure with a multimodal DDPM to refine the encoded latent representations and capture the inter-dependence between two channels.

**Strengths:**

The authors tackled an interesting problem that has not been extensively studied. The writing is clear and easy to follow. The authors provided both qualitative and some quantitative metrics in the evaluation of their framework.

**Weaknesses:**

Although the problem seems interesting, I am not very convinced about the significance and usefulness of generating realistic 3D cells. I would like the authors to provide more backgrounds regarding why they see this problem as important to solve. Also, the experiment section seems a bit brief and weak. The authors compared with several older GAN-based models but lacked comparison with more recent SOTA diffusion-based models. The improvement against MedicalDiffusion in Table 1 looks pretty minor especially for Blebbistatin and Binimetinib groups. The new framework is mostly combining a VQGAN with a multi-channel DDPM in the latent space, and I would like to see at least some sort of ablation study to showcase the importance of having DDPM in the latent space and the usefulness of linking the two modalities together inside the DDPM.

**Questions:**

See above in the "weakness" section.

---

> ### Author Response · Authors · 2024-11-24
> **Reviewer 1**
>
> **The significance of the problem:**
>
> We thank the reviewer for emphasising the importance of clearly articulating the significance and utility of our work. In response, we have revised the Introduction and abstract section to better highlight the relevance of our contributions to the field of pre-clinical drug discovery. Specifically, we now discuss how this work facilitates a notable foundational step towards scalable virtual screening pipelines, enabling the analysis of drug-induced morphological changes at high throughput.
>
> ---
>
> **Brief Experiment Section:**
>
> Thank you for the feedback regarding the need for additional details in the Experiments section. To address this, we have made the following improvements:
>
> 1. Enhanced description of metrics and datasets used.
> 2. Included orthogonal views in the qualitative evaluation figure for a more comprehensive visual analysis.
> 3. Increased the number of generated samples by 4000 per method to ensure a robust cross-validation of our results.
> 4. Incorporated an ablation study into the Experiments section to provide further analysis into the framework’s performance when considering different codebooks/combinations thereof in the latent space.
>
> While we acknowledge the existence of several state-of-the-art diffusion-based generative models, many of these approaches are computationally prohibitive in high-dimensional settings. 3D medical imaging has seen the introduction of a handful of generative models [1][2][3]. Among these, we found MedicalDiffusion [1] to be a notable implementation of high-dimensional diffusion-based modelling, making it a relevant baseline for comparison. We thank the reviewer for this observation, and for future research, we aim to investigate the integration of other diffusion models within our framework to further explore their applicability.
>
> References:
> 1. [https://www.nature.com/articles/s41598-023-34341-2](https://www.nature.com/articles/s41598-023-34341-2)
> 2. [https://www.nature.com/articles/s42256-024-00864-0](https://www.nature.com/articles/s42256-024-00864-0)
> 3. [https://pubmed.ncbi.nlm.nih.gov/35522642/](https://pubmed.ncbi.nlm.nih.gov/35522642/)
>
> ---
>
> **Minor Improvement against MedicalDiffusion in Table 1:**
>
> We thank the reviewer for highlighting the need for additional validation of the quantitative results presented in Table 1. In response, we generated 4000 additional samples per method and conducted cross-validation of the FID and MMD metrics. These additional experiments yielded statistically significant results, providing stronger evidence of the quantitative performance differences between our approach and MedicalDiffusion. This process enhanced the reliability of our evaluation and further substantiated our claims.
>
> ---
>
> **Ablation study of the DDPM and linking the two modalities:**
>
> We are grateful for the reviewer’s recommendation to investigate the interplay between the DDPM components and the linkage of the two modalities in the latent space. This prompted us to include an ablation study at the end of the Experiments section, focusing on the role of the "library of codebooks" in enhancing sample realism. Our results reveal that separating the codebooks enables optimal representation learning, thereby improving synthesis fidelity. Furthermore, the study provided intriguing insights into the relative importance of codebooks in encoding drug-induced phenotypic behaviours. This analysis reinforces the critical role of modality-specific codebooks in achieving biologically realistic 3D cellular structures.

---

> ### Author Response · Authors · 2024-11-26
> **Request reviewer HFNK  to respond**
>
> We kindly request your review of the updates to ensure that the revisions meet your expectations and address your concerns adequately. Your input has been invaluable in improving the quality and clarity of our work, and we look forward to any further suggestions or comments you may have.
>
> Please let us know if you require any additional information or clarification.

---

> ### Author Response · Authors · 2024-12-01
> **Request reviewer HFNK  to respond**
>
> Dear reviewer HFNK, I wanted to follow up on the revised version of our manuscript, which we submitted in response to your insightful feedback. In particular, we addressed your suggestions regarding the significance of our work, expanded the experimental section, included an ablation study, and conducted additional experiments to strengthen our evaluation.
>
> The deadline for final feedback is approaching (December 2nd), and we would greatly appreciate your thoughts on the revisions made.
>
> Thank you once again for your thoughtful comments and for your time in reviewing our work. I look forward to hearing from you.

---

### Author Response · Authors · 2024-11-24
**Thank you to all of the reviewers.**

We sincerely thank the reviewers for their thorough and insightful feedback, which has been invaluable in improving the clarity, quality, and scope of our manuscript. Your constructive comments have helped us better articulate the contributions of our work and refine both the methodology and evaluation.

In the updated manuscript, we have incorporated all suggestions and highlighted the changes in blue text for clarity. We hope these revisions address your concerns and enhance the overall quality of the paper. Thank you for your time and effort in reviewing our work.

---

### Author Response · Authors · 2024-12-02
**Request for reviewer response (a note to area chair)**

Dear Area Chairs,

While we have worked diligently to address the feedback provided by the reviewers and submitted a revised manuscript in advance of the deadline, two of the reviewers (Reviewer HFNK and Reviewer DBvM) have not yet provided feedback on our revisions.

Given that the deadline for final feedback is fast approaching (December 2nd), we are concerned that the lack of response from these reviewers may impede the timely progression of the review process. We greatly value their insights and believe that their comments on our revisions will further enhance the quality of our submission.

If possible, we kindly request your assistance in prompting these reviewers to provide their feedback or advise us on the next steps. Please do not hesitate to let us know if there is any additional information we can provide.

---

> ### Comment · Area_Chair_99cB · 2024-12-02
>
> Dear authors,
>
> I am sorry for the lack of a response and have imparted the relevance of communication on the reviewers. Rest assure that I will judge your submission accordingly and incorporate the lack of response as one relevant factor.

---

### Note · Authors · 2025-02-18

I have read and agree with the venue's withdrawal policy on behalf of myself and my co-authors.

---

### Meta-Review · Area_Chair_99cB · 2024-12-19

**Metareview:**

This submission develops a new framework for generating biologically-plausible cell structures in 3D. Interestingly, the model is also poised towards generating the outcomes resulting from different drug treatments. The resulting samples are quantitatively and qualitatively compared, showing that the proposed model ("Build Your Own Cell," BYOC) results in improved samples as compared to other methods.

This, in addition with several other aspects, is one of the _strengths_ of the paper, to wit:

1. The paper presents a biologically-relevant question and is well-motivated.
2. The presented samples exhibit a higher visual quality than existing methods.
3. The paper is accessible and highly-readable.

Nevertheless, together with the reviewers, I also see some concerns about the _weaknesses_ of the method:

1. The question of whether the resulting samples are already biologically relevant (notice that this is not a contradiction to one of the strengths of the paper, viz. the biologically-relevant research question; my concern—shared with some reviewers—is that the results do not adequately address the question.)
2. The methodological aspects are not sufficiently motivated. While the paper showcases are creative use of machine-learning models to achieve the stated goal, a strong ICLR submission needs to provide insights on the theoretical or on the empirical level. The paper is aiming to provide such empirical insights trough comparisons with existing methods, but these insights could be strengthened by a more thorough biology-driven analysis of the utility of the provided samples.
3. Finally, the evaluation strategy exhibits some issues: While FID and MMD are indeed suitable metrics, the use of [Med3D](https://arxiv.org/abs/1904.00625) results in another component whose influence on the evaluation is hard to assess. Moreover, for the use of MMD, crucial details on parameter choices (like the choice of kernel or its smoothing parameter) are missing. These parameters are known to be critical and the default choices may result in unstable rankings of models; this problem has been discussed in the [context of generative graph neural networks](https://arxiv.org/abs/2106.01098), a particular forté of this AC, but the general point also applies to other models.

As such, I unfortunately have to suggest rejecting the paper in its current form. This decision was not reached lightly, but I believe that a stronger evaluation, in combination with an improved discussion of the biological significance of the findings, would strengthen the paper. For the evaluation part, the authors could, for instance, use [EMD metrics](https://arxiv.org/abs/2210.06978), a staple in 3D point cloud generation, or compare the resulting volumes using DICE/Jaccard (even though these metrics are _not_ invariant to rotations, so potentially, a Procrustes-like alignment might be warranted).

I understand that this is not the desired outcome for the authors, so I want to stress that I believe this work to have strong potential! With a more biology-driven analysis, I could easily imagine this being published in a Nature-like journal as well!

**Additional Comments On Reviewer Discussion:**

Reviewers agreed on the relevance of the problem (`HFNK`, `n4Rv`), appreciated the writing quality (`HFNK`) as well as the method as such (`DBvm`), and parts of the evaluation (`n4Rv`, `wxXn`). Concerns were raised about the generalisation performance, which is somewhat tied to the evaluation issues that I raised above, (`DBvM`, `wxXn`). Initially, some issues about the apparent missing 3D results were raised (`DBvM`), as well as some concerns about the data modality as such (`DBvM`), but these—along with minor issues about accessibility—could be alleviated and addressed by the authors in the rebuttal.

The discussion phase was not marked by engagement from all reviewers, but I want to positively highlight `n4Rv`, whose insightful review resulted in improvements to the text. This was also acknowledged by the reviewer, who raised their score afterwards. Overall, some concerns still remain, and while I believe that the authors adequately responded to some of the points raised by reviewer `HFNK`, who unfortunately did not further engage during the rebuttal, some of the points raised by the reviewer remain unaddressed, like the concern about the significance of the results. This, together with my own assessment of the evaluation issues, forms the basis for my suggestion to the PCs.

---

### Decision · Program_Chairs · 2025-01-22

Reject